# Anti-MAPK Targeted Therapy for Ameloblastoma: Case Report with a Systematic Review

**DOI:** 10.3390/cancers16122174

**Published:** 2024-06-07

**Authors:** Anton Raemy, Laurence May, Nathalie Sala, Manuel Diezi, Maja Beck-Popovic, Martin Broome

**Affiliations:** 1Department of Maxillofacial Surgery, Lausanne University Hospital, 1011 Lausanne, Switzerland; laurence.may@chuv.ch (L.M.); martin.broome@chuv.ch (M.B.); 2Institute of Pathology, Lausanne University Hospital, 1011 Lausanne, Switzerland; nathalie.sala@hopitalvs.ch; 3Department of Paediatric Oncology, Lausanne University Hospital, 1011 Lausanne, Switzerland; manuel.diezi@chuv.ch (M.D.); maja.beck-popovic@chuv.ch (M.B.-P.)

**Keywords:** ameloblastoma, targeted therapy, anti-BRAF/MEK, effectiveness, adverse effect

## Abstract

**Simple Summary:**

Ameloblastoma is a type of tumor that usually forms in the jaw; while typically benign, it grows aggressively and often recurs after treatment. Traditional treatment involves extensive surgery, which can significantly affect a patient’s quality of life. Recent research has focused on a new approach, targeting a specific cellular pathway known as the MAPK pathway, which appears to be involved in the development of these tumors. In this study, we reviewed the outcomes of 23 patients treated with this targeted therapy to assess its safety and effectiveness. The results were promising: most patients experienced significant tumor reduction, and the side effects were generally mild. This suggests that MAPK pathway inhibitors could be a viable alternative to surgery, potentially offering improved outcomes for patients with ameloblastoma, minimizing surgical risks, and preserving quality of life. This advancement could greatly impact the approach to treating this challenging condition.

**Abstract:**

Ameloblastoma, a benign yet aggressive odontogenic tumor known for its recurrence and the severe morbidity from radical surgeries, may benefit from advancements in targeted therapy. We present a case of a 15-year-old girl with ameloblastoma successfully treated with targeted therapy and review the literature with this question: Is anti-MAPK targeted therapy safe and effective for treating ameloblastoma? This systematic review was registered in PROSPERO, adhered to PRISMA guidelines, and searched multiple databases up to December 2023, identifying 13 relevant studies out of 647 records, covering 23 patients treated with MAPK inhibitor therapies. The results were promising as nearly all patients showed a positive treatment response, with four achieving complete radiological remission and others showing substantial reductions in primary, recurrent, and metastatic ameloblastoma sizes. Side effects were mostly mild to moderate. This study presents anti-MAPK therapy as a significant shift from invasive surgical treatments, potentially enhancing life quality and clinical outcomes by offering a less invasive yet effective treatment alternative. This approach could signify a breakthrough in managing this challenging tumor, emphasizing the need for further research into molecular-targeted therapies.

## 1. Introduction

Ameloblastoma is a benign odontogenic tumor that originates from odontogenic epithelium [1]. Although benign, it is a locally aggressive neoplasm that has a strong tendency to recur and may metastasize to distant sites [2]. The global incidence rate of ameloblastoma is 0.92 per 1,000,000 people per year [3] but has strong heterogeneity between countries. There is no sex predilection, and most cases are diagnosed in the third to sixth decade of life [4]. Preoperative diagnostic evaluation involves radiographic imaging and biopsy. There are four main categories of ameloblastoma: conventional, unicystic, extraosseous/peripheral, and metastasizing, each with different histological variants [1].

The standard treatment of ameloblastoma is surgical, to minimize recurrences and restore good function and aesthetics with minimum morbidity in surgical sites (orofacial and donor flap) [5]. The overall recurrence rate for ameloblastoma varies from 11% after radical surgery to 65% after a conservative approach [6]. Due to this high recurrence rate because of the infiltration of tumor cells beyond the radiographic margins of up to 2 to 8 mm [7], the current recommended surgical approach to ameloblastoma (mostly conventional) is radical surgery with adequate margins of safety, which may be difficult to achieve without high morbidity [4].

Only 10–15% of ameloblastoma cases occur in the pediatric population [8,9]. Treatment options for children are also debated as it might be complicated because of three factors: the continuing facial growth and different bone physiology (more cancellous bone, increased bone turnover, and reactive periosteum), the presence of unerupted teeth, and the difficulty in the initial diagnosis [10]. Fearing the potential interruption of facial growth and loss of function, surgeons could choose the most minimal and conservative intervention, which may lead to multiple recurrences. A conservative approach could be used as a temporary procedure in children for the mandible to achieve further growth before carrying out a more appropriate definitive surgery in selected cases. Nevertheless, radical surgery with margins remains the safest treatment option and has the lowest recurrence rate [11]. 

In 2014, three independent studies were published about the genetic profiling of ameloblastomas via DNA sequencing and all three studies showed a high incidence of somatic mutations impacting the mitogen-activated protein kinase (MAPK) that controls cell proliferation [12,13,14]. A specific mutation, BRAF V600E (valine to glutamic acid substitution at amino acid 600 within the BRAF gene on chromosome 7) was reported at high frequencies (63% for Kurppa et al., 46% for Sweeney et al., and 62% for Brown et al.). BRAF is a serine–threonine kinase within the MAPK pathway. This mutation constitutively initiates the mitogen-activated kinase pathway and thus enhances cell proliferation and survival activity in a ligand-independent manner [15]. This mutation is present in numerous neoplasms [16], including melanoma [17], thyroid carcinoma [18], colorectal cancer [19], and low-grade gliomas in children [20,21,22].

Several studies have tried to correlate BRAF mutation with clinicopathological features of ameloblastoma. In a recent systematic review [23], the overall prevalence among ameloblastomas for BRAF V600E mutations was 70.49%, associated with an age of less than 54 years old and without sex predilection. Regarding location, there was an association between the mandible and BRAF mutation. In comparison, for ameloblastomas originating in the maxilla, mutations of the protein Smoothened (SMO) of the Hedgehog pathway, distinctive from the MAPK pathway, are involved [13]. More interestingly, if a BRAF mutation is present, it tends to be exclusive regarding other mutations (SMO, FGFR2, RAS, and MEK) also present in ameloblastoma, making BRAF the most frequent genetic alteration and closely associated with its pathogenesis [14]. In a study of more than 500 ameloblastomas, BRAF V600E cases without additional mutations corresponded to 93.7%; this highlighted why treatment with BRAF inhibitors might be interesting [24]. 

In this study, we present a case of a 15-year-old girl with ameloblastoma, successfully treated with targeted therapy using dabrafenib, and review the literature published up to December 2023 with the following research question: Is anti-MAPK targeted therapy safe and effective for treating ameloblastoma?

## 2. Case Presentation

A 15-year-old Caucasian female was referred in May 2022 to our department of maxillofacial surgery by her orthodontist. Because of an unerupted left second molar, a radiological examination with a dental panoramic radiograph was performed and revealed a voluminous lytic lesion of the left mandibular angle (Figure 1A). The patient was in general good health, had no prior medical or surgical history, and had no complaints at the time.

Extra-oral examination revealed no swelling or neck lymph node enlargement, no deformity, and no paresthesia of the fifth cranial nerve. Intra-oral status was normal except for an impacted second molar. The panoramic radiograph showed a well-defined radiolucent unilobulated expansive lesion from the first molar to mid-ramus, with radiopaque margins/sclerotic borders and unerupted second and third molars.

A provisional diagnosis based on clinical examination and the panoramic radiograph was made as a benign odontogenic lesion. Differential diagnoses considered at the time comprised an odontogenic keratocyst, a dentigerous cyst associated with 37/38, and an unicystic/conventional ameloblastoma. 

An open biopsy and a Cone Beam Computed Tomography (CBCT) were performed in June 2022. The histopathologic examination revealed ameloblastoma (Figure 2A), with positive immunohistostaining for BRAF V600E (clone VE1, Ventana Medical Systems, Figure 2B). Complementary NGS (next-generation sequencing) was performed to analyze the mutational status. DNA was extracted from the paraffin sections containing the tumor and then analyzed on the Ion GeneStudio S5 system (Ion Torrent, Thermo Fischer Scientific, Waltham, MA, USA) using a customized amplicon library (Ion AmpliSeq Custom Cancer Hotspot Panel, version 4.2.4) covering 218 hotspots in 52 genes including BRAF, EGFR, KRAS, NRAS, SMO, and FGFR1-2-3. The analysis confirmed that the only mutation present, BRAF V600E (c.1799T>A), was pathological (score ACGS 5), with a variation allele frequency of 36%. CBCT revealed that the cystic lesion (25.5 mm × 18.8 mm × 24.3 mm) had scalloped margins, an interrupted cortex on the superomedial border, and effraction of the mandibular canal for 14 mm. 

Consideration was given to using the neo-adjuvant MAPK pathway inhibitor therapy, dabrafenib, to reduce the morbidity of the surgery by shrinking the size of the tumor and proceeding to a simple enucleation without injuring the alveolar nerve and the second molar. The patient was then referred to the unit of pediatric hemato-oncology, and treatment with dabrafenib (75 mg bid) was initiated in July 2022. After 2 weeks, the dose was increased to 150 mg bid, with weekly check-ups, including physical examinations, blood work, and urine analysis (platelet count, white blood cell count, absolute neutrophil count, absolute lymphocyte count, hemoglobin, bilirubin, aspartate and alanine aminotransferase, alkaline phosphatase, gamma-glutamyl transferase, amylase, lipase, creatine kinase, phosphate, sodium, glucose, serum creatinine, and estimated glomerular filtration rate). Tolerance was excellent, with grade 1 adverse effects (CTCAE v5.0) of fatigue, dry skin, and arthralgia at the beginning. A CBCT was performed after 2 months of treatment, showing a good response to dabrafenib with progressive mineralization of the cystic lesion. Treatment with the BRAF inhibitor was continued for 1 additional month, with a CBCT in November 2022 to confirm the improvement in cortication with complete ossification around the alveolar nerve and sufficient reduction of the lesion to proceed to an enucleation without osteotomies and loss of bone length. After discussion, the patient and her mother elected to proceed to surgical intervention, and written consent was obtained from the patient and her mother. The surgery took place in mid-November, after 102 days of dabrafenib. During surgery, curettage of the cavity was not necessary as new bone of perfect quality had been formed and a simple enucleation of the residual tumor with the extraction of the third molar took place; the second molar remained entirely in the newly formed bone. Biopsies of the newly formed bone and enucleation of the residual lesion were analyzed. The bone showed no evidence of BRAF mutation, nor alteration, and remained of good histological quality. Residual epithelial tissue (16 mm × 13 mm × 0.9 mm) confirmed an ameloblastoma of 6 mm × 6 mm associated with signs of post-therapeutic regression in the form of fibro-inflammatory changes (Figure 2C,D) and confirmed the mutation of BRAF V600E (IHC and NGS) with an allelic frequency of up to 1%. Surgery and the postoperative course remained uncomplicated. Serial follow-up and postoperative CBCT at 3, 12, and 24 months showed appropriate healing of the operative site and no sign of recurrence, with an adequate eruption of the second molar and complete anatomical restoration of the cortical (Figure 1B,C). Ideally, serial follow-up for the case should be for at least 60 months, probably up to 120.

## 3. Materials and Methods

### 3.1. Research Question, Protocol, and Eligibility Criteria

The report presentation followed Preferred Reporting Items for Systematic Review and Meta-Analyses (PRISMA) guidelines from the screening protocol to the final analysis [25]. The protocol has been registered in the PROSPERO database (CRD42024484742). The following research question was formulated: Is anti-MAPK targeted therapy safe and effective for treating ameloblastoma?

The inclusion criteria for the studies to be considered in this systematic review were as follows: (1) any clinical research evaluating anti-BRAF/MEK or targeted therapies (MAPK pathway) for the treatment of ameloblastoma, ameloblastoma recurrences, and their transformation (ameloblastic carcinoma, fibroblastic ameloblastoma, and metastasizing ameloblastoma), (2) studies related to the use of anti-BRAF/MEK or targeted therapies only in in vivo human studies, (3) English-language articles. The exclusion criteria were as follows: (1) studies of in vitro or animal use of targeted therapies, (2) abstract-only papers as preceding papers and conference, editorial, and author response theses and books, (3) studies reported as review papers, books, practice guidelines, letters, editorials, or commentaries.

### 3.2. Data Sources, Search Strategy, and Study Selection

A comprehensive electronic search of the PubMed, Google Scholar, Embase, Web of Science, and Medline databases without any restrictions in the year of publication until December 2023 was conducted. The main keywords according to the PICO tool [26] were selected as follows: ameloblastoma (Population), anti-MAPK targeted therapy (Intervention), Comparison (not applicable), adverse effects, and tumor response to treatment (Outcome). The search strategy involved a combination of keyword concepts and medical subject heading (MeSH) terminology. The Booleans and/or were used along with the following terms: “ameloblastoma”, “dabrafenib”, “trametinib”, “vemurafenib”, “B-RAF inhibitor”, “MEK inhibitor”, “proto-oncogene Proteins B-raf”, “MEK”, “targeted therapy”, and “anti-MAPK pathway”.

The selection of studies was carried out by two independent authors (AR and LM) by screening the titles and abstracts of studies to identify articles potentially meeting inclusion criteria. The full texts of these potentially eligible trials, as well as of those abstracts that did not provide sufficient information to allow a decision regarding inclusion or exclusion, were retrieved and assessed by the same review authors (AR and LM). Any discrepancy was checked by a third author (MB) who started analysis from initial screening until the assessment of bias. Data were extracted by one author (AR), and checked by the second (LM), into a table specifically designed for this review. Discrepancies were resolved in the same way as above by another author (MB). Relevant information was gathered in a table as follows: study (author, year), age, sex, initial diagnosis and medical history, location, symptoms, mutational status (IHC, SNG), treatment type and modality, treatment duration, radiological response, pathological response, symptoms response, adverse effect CTCAE, treatment modification, maximum follow up, and recurrence.

### 3.3. Quality Assessment and Data Synthesis

The risk of bias in the selected studies was assessed using the Joanna Briggs Institute (JBI) critical appraisal checklist for case reports [27] and case series [28], which classifies case reports and case series in terms of overall completeness, scientific rigor, methodological quality, and data analysis. Studies that did not satisfy the JBI checklist were excluded. Quality assessment was done by two authors (AR and LM) independently. Given the descriptive nature of this review, data synthesis was reported in a qualitative form with both primary and secondary outcomes recorded in tables. Descriptive statistics were used to report demographics and clinical characteristics, with means and standard deviation for continuous variables, and frequencies or percentages for dichotomous variables. Because of a small number of cases present in the literature with several variables and because the outcome measures were of a heterogeneous nature, quantitative analysis could not be conducted.

## 4. Results

### 4.1. Data Search Results and Quality Assessment

In total, 647 abstracts and titles were obtained through electronic database research (Figure 3). After removing duplicates, 293 articles were included for the screening by reading titles and abstracts. Consequently, 259 articles were excluded as they did not fulfil the inclusion criteria and 34 articles were screened in detail. Finally, 21 articles were excluded as they did not meet the study aims (reason 1, because of mostly demographic analysis with BRAF, and reason 2, because of samples taken from another study [29,30]. The 13 remaining articles were included in this systematic review and were evaluated for risk of bias according to JBI tools for case reports as 11/13 were case reports or case series (Figure 4). The two studies left [31,32] were also included in this review despite the lack of demographic information regarding the patients as they are two clinical trials from NCI-MATCH that include one patient each with ameloblastoma treated with targeted therapy.

### 4.2. Study Characteristics

Ten case reports, one case series, and two clinical trials were included in this review, with a total of 23 patients treated with a MAPK pathway inhibitor for ameloblastoma or its variants. The main findings regarding demographics and treatment modalities are summarized in Table 1 and Table 2, respectively. Out of the 23 patients, 15 were males (71.5%) and six were females (28.5%), with a mean age of 35.19 years (range from 10 to 85), a standard deviation of 25.38, and two missing data for these demographics. Out of the 22 patients diagnosed (one missing data), 13 had primary ameloblastoma (59.1%), six had recurrent ameloblastoma (27.3%), and four had metastasizing ameloblastoma (18.2%), with a single case presenting as both locally recurrent and metastasizing [33]. All patients were screened for mutational status, with a large prevalence of BRAF V600E mutation (87%), one case of NRAS mutation (4.3%) [32], and two cases of FGFR2 mutation (8.7%) [34,35], all diagnosed by NGS/PCR. Regarding the treatment, the majority used dabrafenib, either in combination with trametinib (39.1%) or alone (30.4%). In two situations, vemurafenib was prescribed (8.7%) [36,37], and single cases (4.35%) of trametinib [38], binimetinib [32], lenvatinib [35], and erdafitinib [34], along with a combination of vemurafenib and cobimetinib [39], were listed (Figure 5). When using dabrafenib with trametinib, the dose was 150 mg BID and 2 mg QD, respectively. For dabrafenib alone, the dose was usually 4 mg/kg/d in pediatrics, or 150 mg BID. The modality of treatment was 52.2% neo-adjuvant use (followed by surgery) and 47.8% exclusive use. Two rechallenges were recorded [33,34], and two cases of neo-adjuvant and adjuvant use [39,40]. The mean duration of treatment, in total and with the exclusion of cases of treatment discontinuation [32,40,41], rechallenges [33,34], and unresponsive treatment [38], was 13.43 months (ranging from 4 months to 35 months). The mean duration for the exclusive modality of treatment was 14.25 months and the mean duration for the neo-adjuvant modality was 12.11 months (with previous exclusion).

### 4.3. Primary Outcome

Concerning the tumor response to treatment, the main findings are reported in Table 3. Regarding the radiological response, from the 23 cases analyzed, four demonstrated a complete response (17.4%) including two cases of metastasizing ameloblastoma in the lungs [33,42] and one rechallenge [34], 18 showed a partial response (78.3%), and three presented stable disease (13.1%), with one case of both a complete response for lung metastasis and a locally partial response, and one case of a partial response initially and then a near complete response at rechallenge. Regarding the pathological response, out of the 13 neo-adjuvant cases where surgery was performed enabling an anatomopathological examination, one case of a complete response meaning no ameloblastic cells or amelobastoma features (7.7%), four cases of near-complete responses (30.7%), seven cases of partial responses (53.8%), and one case of stable disease (7.7%) were reported. Considering the effect on previous symptoms reported by patients, out of the nine cases mentioning a change, eight described a complete response, meaning complete resolution of previously mentioned symptoms (88.9%), and one described no improvement (11.1%). The mean time to a complete response for symptoms was 101 days with a range from 2 days to 240 days. The mean follow-up time was 26.71 months. Concerning the recurrence, out of the 21 cases included (exclusion of daws and one missing data), 18 did not recur (85.7%) and three did recur (14.3%) at a mean interval of 19.3 months after the initial treatment (exclusive or neo-adjuvant with surgery), and all were rechallenged with the same targeted therapy used before. All three rechallenges showed a partial or near-complete radiological response and treatment is still ongoing.

### 4.4. Secondary Outcome

Regarding the adverse effects of targeted therapies, the main findings are described in Table 4. From the 23 patients, five missing data were reported, 16/18 (88.9%) reported an adverse event, and 2/18 (11.1%) reported no side effects from the therapy. Out of these 16 cases, there were two cases of grade 1 (6.25%), 10 cases of grade 1–2 (62.5%), two cases of grade 2 (12.5%), one case of grade 3 (6.25%), and one unspecified grade (6.25%). Most of the adverse events (60%) occurred in the skin and subcutaneous tissue or were of a general type (e.g., fever, fatigue) according to the CTCAE v5.0. Of the 16 adverse events, six required no special treatment (37.5%), 10 needed a treatment break or a dose reduction (62.5%), and three called for discontinuation (18.75%). Of these three discontinuations, only one was properly due to a grade 3 adverse event according to Grynberg et al. [40]; the two others were due to a lack of compliance.

## 5. Discussion

Ameloblastoma is generally a benign odontogenic tumor; however, it is locally invasive. Standard treatment nowadays is mainly based on surgery, either using a conservative approach or a radical one. Based on the high impact on recurrence of the conservative approach [6,44], radical surgery is the gold standard treatment, even though it comes with a potential impact on quality of life and morbidity. Recurrent ameloblastoma, just like primary ameloblastoma, still represents a treatment challenge as it usually occurs near the base of the skull or neurovascular structures, where chemotherapy and radiation fail to reach [45,46]. However, with the recent emphasis and confirmation of the involvement of the MAPK pathway in the pathogenesis of ameloblastoma [47], and the rising use of specific inhibitors known as anti-MAPK targeted therapies [48], a new era of promising tumor-specific treatment has begun. This case report and review of the literature aimed to demonstrate that the use of a targeted therapy for ameloblastoma is safe and efficient.

This study reported on 13 articles with 23 patients diagnosed with ameloblastoma or its variants and treated with targeted therapy focusing only on the MAPK pathway. BRAF V600E represents most of the mutated genes in that pathway for this review. Nevertheless, two cases of mutation in FGFR2 were reported, associated with the second most relevant mutation in ameloblastoma, the somatic mutation of the Smoothened (SMO) gene in the sonic hedgehog (SHH) pathway [49], as well as a single case of NRAS mutation. This is consistent with the mutations reported in the literature [50], as BRAF is the most studied protein in ameloblastoma pathogenesis [51]. The MAPK pathway seems to be the keystone to targeting.

A review of treatment outcomes with the BRAF inhibitor Dabrafenib alone or in association with the MEK inhibitor Trametinib shows four complete responses, a majority of partial responses, and one case of stable disease that was presented as unresponsive. This one case considered as unresponsive was a pediatric patient with primary ameloblastoma treated with a monotherapy of the MEK inhibitor trametinib for institutional preferences as the mutational status was BRAF V600E-positive. The patient was finally treated with radical surgery after eight weeks of treatment with no radiological change [38]. After surgery, the histopathological analysis of the postoperative specimen confirmed no squamous differentiation and no necrosis. Most of the partial responses were with the neo-adjuvant treatment modality. Grynberg et al. [40] reported eleven cases of this upfront modality, with one complete response to dabrafenib combined with trametinib allowing the authors to skip surgery because of full mandibular regeneration and re-ossification. In the ten other cases, including five pediatric patients, partial responses were achieved and were followed by organ-preserving surgeries, meaning that dabrafenib alone or in combination with trametinib was given until a radiological response allowed conservative surgery. The median time for this criterion was 12 months. Pathological response was assessed on the operative specimen with four near complete responses, meaning the replacement of tumor tissue by fibrosis and newly formed bone with characteristics of ameloblastoma lost upon microscopical analysis. Regarding the three other complete responses achieved in this review, two concerned metastasizing ameloblastoma in the lung. The first case is a recurrence from Kaye et al. published by Ambramson et al.; a complete response in the lungs for metastasizing recurrent ameloblastoma being rechallenged with the same inhibitor therapies. This finding is consistent with the recurrence of melanoma after achieving a complete response followed by a treatment break [52,53]. Another complete response on metastasizing ameloblastoma in the lungs was achieved by Brunet et al. and on a rechallenge from Lawson-Michoud et al., with a near complete response locally after a recurrence of a partial response 7 years ago.

Three cases of recurrences were spotted. One of them was on the case series of Grynberg et al. [40], after 13 months of medication and 8 months after initial surgery. The recurrence was displayed as a soft tissue focus within the mandibular bone that was re-operated and inhibitor treatment was renewed as adjuvant therapy, which is still ongoing. The other two cases were discussed above.

There was no instance of SMO-targeted medication for ameloblastoma in this literature review. This mutation has been reported to have a substantially greater effect on the maxilla rather than the mandible [13] and to be associated with resistance to the SMO inhibitor therapy vismodegib [54]. The two cases of SMO mutation were associated with a mutated variant of FGFR2 and were successfully treated with their inhibitor, erdafitinib or lenvatinib. A single case of NRAS mutation was registered and treated with its inhibitor, binimetinib, and achieved a partial response that is still ongoing. This case was part of the NCI-MATCH clinical trial [55] and was the only durable PR reported.

Considering the toxicity of these given therapies, there was only one treatment discontinuation on grade 3 toxicity with dual inhibitor BRAF/MEK therapy [40]. The two other treatment discontinuations could not be assigned to treatment toxicity and were due to poor compliance. Most of the adverse effects were rash, arthralgia, pyrexia, headache, asthenia, and nausea [56], and were in accordance with our report. Even in cases with pediatric patients, the patients showed generally good tolerance, which reinforces the treatment’s potential as an alternative therapy to surgery. Two-thirds of the patients in our review needed a simple treatment adjustment, either a short break or a temporary dose reduction, and the remaining third carried on under optimal treatment doses without any change. However, using anti-BRAF therapy alone might come with downsides such as paradoxical activation of the MAPK pathway and the consequent development of resistance-acquired skin tumors (keratoacanthomas and cutaneous squamous cell carcinomas) [57], yet none of the 13 articles in this review mentioned it. Adding an anti-MEK to dabrafenib seems to reduce the rate of secondary skin cancer in melanoma patients [58]. In addition, there is no consensus for the follow-up time of ameloblastomas after anti-MAPK targeted therapy. Therefore, long-term effects should be studied in the future.

These recent advances in the molecular signaling pathways associated with the pathogenesis of ameloblastoma have led to the use of targeted therapies for the management of this tumor. Several MAPK-specific drugs selectively inhibit the function of mutated BRAF and others among the MAPK pathways to stop the dysregulated proliferation and differentiation of ameloblastic cells. Neo-adjuvant treatment with anti-BRAF, resulting in impressive tumor regression, has enabled non-mutilating complete surgical removal and mandible preservation as well as conservation of teeth in growing patients, as we demonstrated in our case report. While the reduction in tumor size was witnessed through radiological monitoring, reconstruction and integrity of the original cortical bone structure of the mandible were observed. It underlines the utility of close clinical and radiological monitoring, as the challenge is to obtain maximal responses in order to plan surgery right on time. Ongoing clinical studies are highlighting the surgical improvement in outcomes for high-risk melanoma with inductive anti-BRAF therapy [59]. The key benefit of neo-adjuvant molecular targeted therapies is that they can significantly reduce the surgical morbidity seen in radical surgery with either primary or recurrent ameloblastoma. This also emphasizes the necessity of systematic gene profiling for each tumor, allowing for the identification of potential targeted therapies with a multidisciplinary team.

Our findings regarding the exclusive modality treatment of anti-MAPK drugs seem to be appropriate for advanced disease with high-risk surgeries and where there are limited options, often after multiple conservative attempts and recurrences. The clinical outcome has been improved for patients with unresectable ameloblastoma treated with targeted therapy. As there is no standard therapy for managing metastasizing ameloblastoma, we believe MAPK-specific inhibitors appear very promising. Considering the exclusivity of the BRAF mutation in the genetic profiling of ameloblastoma and the importance of MAPK pathway activation for this disease, MAPK inhibitors should be evaluated prospectively as a novel cure for this tumor.

There are several limitations to our report. First, the sample size of the systematic review is small, and the origin of the data was nearly all retrospective case reports. This could generate potential selection bias, and some information is bound to be missing. Second, there is a lack of homogeneity as there are primary ameloblastoma from different histological types, recurrent, metastasizing, and different modalities of treatment, as well as no uniformity of treatment types and regiments. Despite this, all case reports have the potential to detect novelties, generate new hypotheses, and are useful for the investigation of rare diseases. Furthermore, quality assessment was generally good.

## 6. Conclusions

Anti-MAPK targeted therapy seems to be safe and efficient in treating ameloblastoma that is primary, recurrent, or metastasizing. It allows for a clinical, radiological, pathological, and symptom-relieving response for this rare but potentially devastating disease. There is now a need to confirm the retrospective data within a prospective study on the use of targeted therapy for ameloblastoma, and considering the rarity of this disease, to plan it in a collaborative setting between institutions, nationally or even internationally, with the aim of improving outcomes and quality of life for these patients.

## Figures and Tables

**Figure 1 cancers-16-02174-f001:**
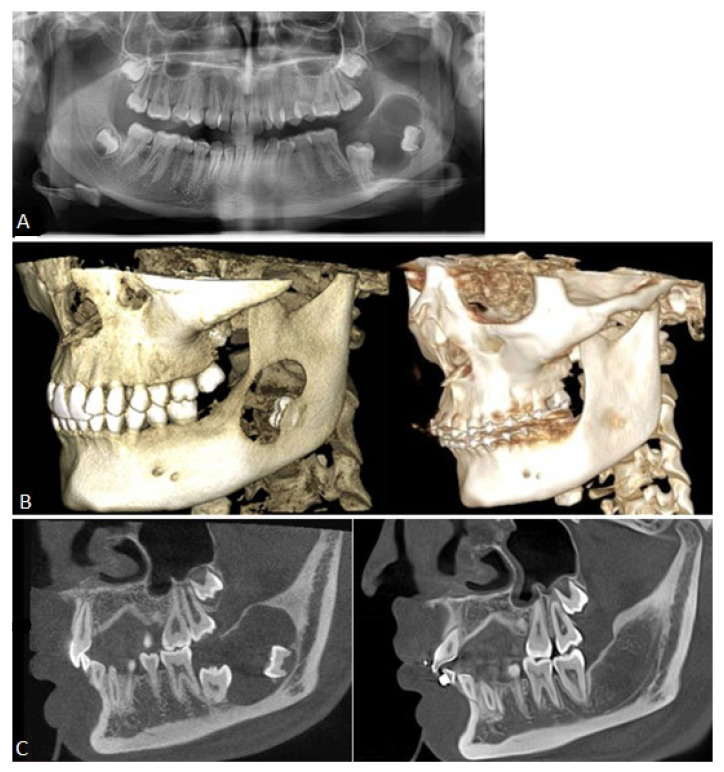
Different radiographic images before and after treatment, showing the evolution of the lesion. (**A**): Panoramic X-rays before treatment with a cystic lesion, unerupted second molar, and interrupted cranial cortical. (**B**): A 3D reconstruction of a CBCT pre- and post-treatment with complete ossification of the interrupted cortical border of the mandible. (**C**): CBCT in a modified sagittal view pre- and post-treatment. To note, the complete eruption and occlusion of the second molar.

**Figure 2 cancers-16-02174-f002:**
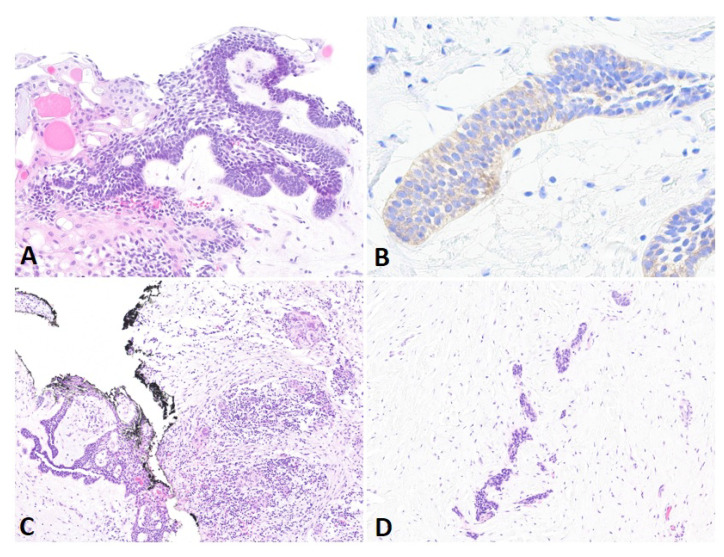
Different histological analysis. (**A**): Histological analysis of the initial biopsy shows epithelial strands of basal cells with palisaded and hyperchromatic nuclei, and centrally located stellate reticulum-like cells with acanthomatous changes (H&E, 20×). (**B**): BRAF immunostain shows moderate cytoplasmic positivity in tumoral cells (40×). (**C**,**D**): Enucleation after anti-BRAF therapy shows residual tumor admixed with inflammatory cells, including lymphocytes and giant multinucleated cells (**left**), and surrounded by large areas of fibrosis (**right**) (H&E, 20×).

**Figure 3 cancers-16-02174-f003:**
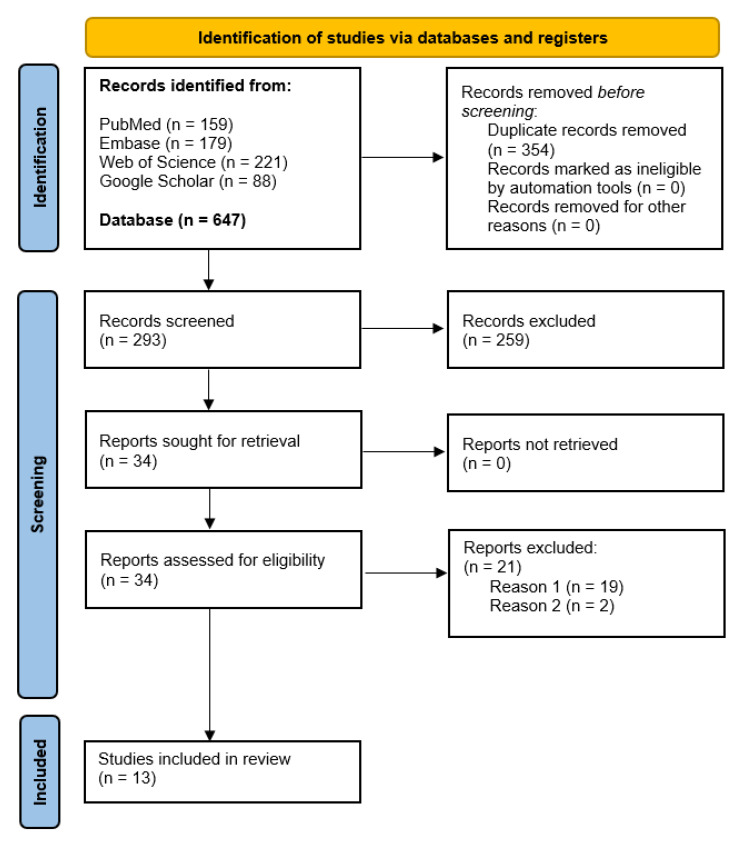
Moher D. et al. [25] for the Preferred reporting guideline in systematic reviews and meta-analysis (PRISMA) flow diagram.

**Figure 4 cancers-16-02174-f004:**
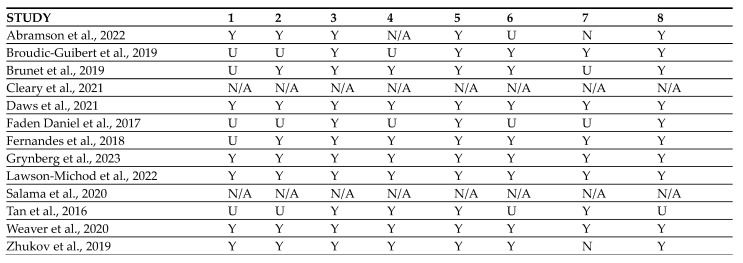
Risk of Bias assessment of the included studies [31,32,33,34,35,36,37,38,39,40,41,42,43]. Domains of the Risk of Bias Assessment Tool. (1) Were the patient’s demographic characteristics clearly described? (2) Was the patient’s history clearly described and presented as a timeline? (3) Was the current clinical condition of the patient on presentation clearly described? (4) Were diagnostic tests or assessment methods and the results clearly described? (5) Was the intervention(s) or treatment procedure(s) clearly described? (6) Was the post-intervention clinical condition clearly described? (7) Were adverse events (harms) or unanticipated events identified and described? (8) Does the case report provide takeaway lessons? (Y) Yes, (N) No, (U) Unclear or (N/A) Not/Applicable.

**Figure 5 cancers-16-02174-f005:**
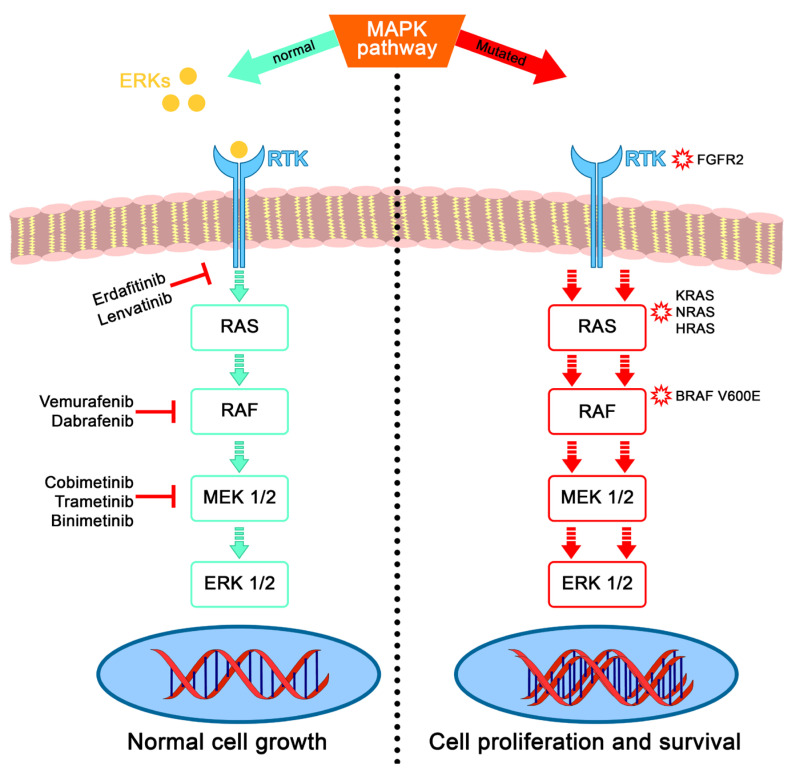
Schema of the mitogen-activated protein kinase (MAPK) pathway from the outer membrane to the nucleus. On the (**left**), its normal activation by Extracellular signal-regulated Kinases called ERK (FGF, EGF, etc.) binding to receptor tyrosine kinase RTK (FGFR, EGFR, etc.) and thus activating the cascade signaling through RAS–RAF–MEK–ERK, finally regulating cellular biological function. Without extracellular signals activating the phosphorylation of RAS, the pathway is switched off. Few examples of targeted therapies. On the (**right**), the different mutated genes of the pathway cause its constitutive activation without external stimuli, leading to cell proliferation. Mutations in the RAS gene (NRAS, KRAS, and HRAS), in the FGFR gene (FGFR2), and in the RAF gene (BRAF) are the most common.

**Table 1 cancers-16-02174-t001:** Demographics (M male, F female, NS non-specified, IHC Immunohistochemistry, NGS next-generation sequencing, PCR polymerase chain reaction, BID twice a day, QD once a day, CR complete response, NCR near-complete response, PR partial response, SD stable disease, N no, TTT treatment).

Study	Age	Sex	Diagnosis	Location	Mutational Status
Abramson et al. [33] 2022	47	M	Ameloblastoma, recurrent and metastatic (2 times)	Left mandible, bilateral lung metastasis	BRAF V600E, IHC + NGS
Broudic-Guibert et al. [36] 2019	33	F	Ameloblastoma plexiform, lung metastases	Left mandible, bilateral lung metastasis	BRAF V600E
Brunet et al. [42] 2019	26	F	Ameloblastoma, lung metastases	Right mandible, bilateral lung metastasis	BRAF V600E, NGS
Cleary et al. [32] 2021	NS	NS	Ameloblastoma, lung metastases	NS, bilateral lung metastasis	NRAS Q61R, NGS
Daws et al. [38] 2021	13	F	Ameloblastoma, primary	Right mandible	BRAF V600E, PCR
Fernandes et al. [37] 2018	29	F	Ameloblastoma, recurrent	Left mandibular ascending branch, extension to cavernous sinus and orbital fissure	BRAF V600E, PCR
Faden Daniel et al. [43] 2017	83	F	Ameloblastoma, recurrent	Right mandibular body	BRAF V600E
Grynberg et al. [40] 2023	15	M	Ameloblastoma unicystic mural type, primary	Right mandibular angle	BRAF V600E, IHC + NGS
	13	M	Ameloblastoma unicystic mural type, primary	Right mandibular angle	BRAF V600E, IHC + NGS
	11	M	Ameloblastoma unicystic mural type, primary	Left mandibular angle	BRAF V600E, IHC + NGS
	15	M	Ameloblastoma unicystic mural type, primary	Right mandibular angle	BRAF V600E, NGS
	21	M	Ameloblastoma unicystic mural type, primary	Left mandibular angle	BRAF V600E, NGS
	83	M	Ameloblastoma conventional, primary	Mandibular symphysis	BRAF V600E, NGS
	32	M	Ameloblastoma unicystic mural type, primary	Right mandibular body	BRAF V600E, NGS
	15	F	Ameloblastoma unicystic mural type (borderline conventional, primary	Right mandible	BRAF V600E, NGS
	63	M	Ameloblastoma conventional, primary	Right mandible	BRAF V600E, NGS
	19	M	Ameloblastoma unicystic mural type, primary	Left mandibular angle	BRAF V600E, NGS
	25	M	Ameloblastoma conventional, primary	Right mandibular angle	BRAF V600E, NGS
Lawson-Michod et al. [34] 2022	40	M	Conventional ameloblastoma, recurrent	Right maxillary sinus, extension to lateral orbit, middle cranial fossa, temporal lobe	PD-L1 CPS 2%, FGFR2 V395D, SMO W535L
Salama et al. [31] 2020	NS	NS	Ameloblastoma, NS	Mandible	BRAF V600E, NGS
Tan et al. [41] 2016	85	M	Ameloblastoma follicular and plexiform, recurrent	Left mandibular angle	BRAF V600E, PCR + IHC
Weaver et al. [35] 2020	62	M	Ameloblastoma plexiform, primary	right maxillary sinus	FGFR2 Y375C, SMO L412F, PALB2 H786Y
Zhukov et al. [39] 2019	10	M	Ameloblastic fibroma, recurrent	Right mandible, extension to base skull	BRAF V600E

**Table 2 cancers-16-02174-t002:** Treatment modalities.

Study	N°	Medication	Dosis	Duration	Treatment Modality
Abramson et al. [33] 2022	1	Dabrafenib Trametinib	150 mg BID 2 mg QD	4 y, stop 3 y, rechallenge 16 m (ongoing)	Rechallenge, exclusive
Broudic-Guibert et al. [36] 2019	1	Vemurafenib	960 mg BID, reduce to 720 mg BID, 480 mg BID	26 m (ongoing)	exclusive
Brunet et al. [42] 2019	1	Dabrafenib Trametinib	150 mg BID 2 mg QD	30 w (ongoing)	exclusive
Cleary et al. [32] 2021	1	Binimetinib	45 mg BID	26 m	exclusive
Daws et al. [38] 2021	1	Trametinib	1.5 mg QD, reduced to 1 mg QD	2 w, 6 w	exclusive
Fernandes et al. [37] 2018	1	Vemurafenib	960 mg BID	12 m (ongoing)	exclusive
Faden Daniel et al. [43] 2017	1	dabrafenib	75 mg BID, reduced to 32.5 mg BID	12 m (ongoing)	exclusive
Grynberg et al. [40] 2023	1	Dabrafenib	4.5 mg/kg/d	20 m	Neo-adjuvant
	2	Dabrafenib	4.5 mg/kg/d	18 m	Neo-adjuvant
	3	Dabrafenib	4.5 mg/kg/d	12 m	Neo-adjuvant
	4	Dabrafenib	4.5 mg/kg/d	6 m	Neo-adjuvant
	5	Dabrafenib Trametinib	150 mg BID 2 mg QD	13 m	Neo-adjuvant and adjuvant
	6	Dabrafenib Trametinib	150 mg BID 2 mg QD	4 m	Neo-adjuvant
	7	Dabrafenib Trametinib	150 mg BID 2 mg QD	16 m	Neo-adjuvant
	8	Dabrafenib	4.5 mg/kg/d	12 m	Neo-adjuvant
	9	Dabrafenib Trametinib	150 mg BID 2 mg QD	8 m	Neo-adjuvant
	10	Dabrafenib Trametinib	150 mg BID 2 mg QD	3 m	Neo-adjuvant
	11	Dabrafenib Trametinib	150 mg BID 2 mg QD	4 m	Exclusive
Lawson-Michod et al. [34] 2022	1	Pembrolizumab Erdafitinib	NS 8 mg QD	12 w 12 m, stop 14 m, rechallenge 12 w (ongoing)	Rechallenge, exclusive
Salama et al. [31] 2020	1	Dabrafenib Trametinib	150 mg BID 2 mg QD	35 m (ongoing)	Exclusive
Tan et al. [41] 2016	1	Dabrafenib	150 mg BID	73 d	Neo-adjuvant
Weaver et al. [35] 2020	1	Lenvatinib	24 mg QD, reduced to 20 mg, reduced to 20–10 mg, reduced to 14 mg and again 20 mg	1 m, 1 m, 13 m in total (ongoing)	Exclusive
Zhukov et al. [39] 2019	1	Vemurafenib Cobimetinib	960 mg BID 60 mg QD	18 m	Neo-adjuvant and adjuvant

**Table 3 cancers-16-02174-t003:** Tumor response to treatment.

Study	N°	Radiological Response	Pathological Response	Effect on Symptoms	Follow-Up	Recurrence
Abramson et al. [33] 2022	1	CR for lung metastasis, PR locally		CR after 2 w	7 y	Yes (treated with rechallenge)
Broudic-Guibert et al. [36] 2019	1	PR, 30% decrease in diameter of lung meta after 3.5 m (RECIST)		CR after 4 m	26 m	N
Brunet et al. [42] 2019	1	CR (RECIST + PERCIST)		CR after 30 w	30 w	N
Cleary et al. [32] 2021	1	PR		NS	26 m	N
Daws et al. [38] 2021	1	SD	No necrosis, no squamous differentiation, no architectural change	SD	3 m	/
Fernandes et al. [37] 2018	1	PR		CR after 2 w	12 m	N
Faden Daniel et al. [43] 2017	1	PR, 75% volume reduction after 8 months		CR at 8 m	12 m	N
Grynberg et al. [40] 2023	1	PR	Near Complete Response	NS	58 m	N
	2	PR	PR	NS	51 m	N
	3	PR	PR	NS	42 m	N
	4	PR	NCR	NS	17 m	N
	5	PR	PR	NS	33 m	Yes (treated with re-operation followed by adjuvant rechallenge)
	6	PR	PR	NS	23 m	N
	7	PR	NCR	NS	26 m	N
	8	PR	NCR	NS	30 m	N
	9	PR	PR	NS	19 m	N
	10	PR	PR	NS	15 m	N
	11	CR		NS	15 m	N
Lawson-Michod et al. [34] 2022	1	PR, NCR at rechallenge		CR after 4w	29 m	Yes (treated with rechallenge)
Salama et al. [31] 2020	1	SD		NS	35 m	NS
Tan et al. [41] 2016	1	SD	Squamous cell differentiation. Less than 10% of ameloblastic cells	NS	1 y	N
Weaver et al. [35] 2020	1	PR, 40% volume reduction after 6 m (RECIST)		CR et 6m	13 m	N
Zhukov et al. [39] 2019	1	PR, 79% volume reduction after 1 m	No BRAF sensitive allele-specific PCR, CR	CR after 2d	18 m	N

**Table 4 cancers-16-02174-t004:** Adverse effects of targeted therapies.

Study	N°	Medication	Dose	CTCAE (Grade, Symptoms)		TTT Modification
Abramson et al. [33] 2022	1	Dabrafenib Trametinib	150 mg BID 2 mg QD	NS		
Broudic-Guibert et al. [36] 2019	1	Vemurafenib	960 mg BID, reduce to 720 mg BID, 480 mg BID	Grade 1–2	Arthralgia, nausea, rash	Dose reduction (50%)
Brunet et al. [42] 2019	1	Dabrafenib Trametinib	150 mg BID 2 mg QD	NS		
Cleary et al. [32] 2021	1	binimetinib	45 mg BID	Grade 2	Myalgia	Discontinuation
Daws et al. [38] 2021	1	trametinib	1.5 mg QD, reduced to 1 mg QD	(Grade 1–2)	Modeste pustular facial acne	TTT held 1w, dose reduction
Fernandes et al. [37] 2018	1	vemurafenib	960 mg BID	Grade 1	Anorexia, nausea, fatigue	none
Faden Daniel et al. [43] 2017	1	dabrafenib	75 mg BID, reduced to 32.5 mg BID	NS		
Grynberg et al. [40] 2023	1	Dabrafenib	4.5 mg/kg/d	Grade 2 Grade 1	Fever Curly hair	TTT held
	2	Dabrafenib	4.5 mg/kg/d	Grade 1	Fever, rash acneiform	none
	3	Dabrafenib	4.5 mg/kg/d	Grade 2 Grade 1	Erythema nodosum Folliculitis	TTT held
	7	Dabrafenib Trametinib	150 mg BID 2 mg QD	Grade 1–2	Fever, weakness, arthralgia	TTT held and dose reduction
	9	Dabrafenib Trametinib	150 mg BID 2 mg QD	Grade 1–2	Fever, myalgia, aphthous ulcers	TTT held and dose reduction
	10	Dabrafenib Trametinib	150 mg BID 2 mg QD	Grade 3	Hepatitis	discontinuation
	unspecified	unspecified	/	Grade 1–2 (3 patients)	unspecified	none
Lawson-Michod et al. [34] 2022	1	pembrolizumab erdafitinib	NS 8 mg QD	Grade 2	Myalgia, fatigue	TTT held
Salama et al. [31] 2020	1	dabrafenib trametinib	150 mg BID 2 mg QD	NS		
Tan et al. [41] 2016	1	dabrafenib	150 mg BID		Fatigue, actinic keratoses, voice change	discontinuation
Weaver et al. [35] 2020	1	lenvatinib	24 mg QD, reduced to 20 mg, reduced to 20–10 mg, reduced to 14 mg and again 20 mg	Grade 2 Grade 1	HTA, hypothyroidism, weight loss diarrhea	none
Zhukov et al. [39] 2019	1	vemurafenib cobimetinib	960 mg BID 60 mg QD	NS		

## Data Availability

The original contributions presented in the study are included in the article, further inquiries can be directed to the corresponding author.

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
