# Peer review of "Anti-MAPK Targeted Therapy for Ameloblastoma: Case Report with a Systematic Review"

_cancers, 2024, doi:10.3390/cancers16122174_

Round 1
Reviewer 1 Report
Comments and Suggestions for Authors
This is an interesting paper, worthy of publication. I have some concerns regarding the paper.
1. Line 10 "affect a patients
2. Line 37 I suggest "Although benign, it is a locally aggressive neoplasm that has a strong tendency to recur and may the tumour may metastasize to distant sites." a new reference may be required here as some of the cases in the systemic review have metastatic disease. A comment regarding the histology of metastatic case
3. Line 145, Ideally serial follow up for the reported case should be at least 60 months and probably more up to 120. The authors should mention this in the article.
4. Line 317, replace the word indeed with "generally" the article reports cases of metastatic ameloblastoma which we could accept as not benign
Comments on the Quality of English Language
There are some english language corrections required
1. Line 10 "affect a patient's
2. Line 97 delete"but" replace with "except for"
3. Line 115 scalloped margins, change cortical to cortex
4. line 147 change cortical to cortex
5. Line 398 replace integral with "integrity"
6. Line 405 replace to with "for"
7. Line 426 replace to obtain with "for"
Author Response
Dear Reviewer,
thank you very much for your relevant review and your time taken to review my article. Also, thanks a lot for your positive and supporting comment, it is much appreciated.
- Comment and Suggestions for Authors
- Line 10 "affect a patient's : OK done
- Line 37 I suggest "Although benign, it is a locally aggressive neoplasm that has a strong tendency to recur and may the tumour may metastasize to distant sites." a new reference may be required here as some of the cases in the systemic review have metastatic disease. A comment regarding the histology of metastatic case: thank you for pointing this out. metastasizing ameloblastoma remain rare but appear several time in this report indeed. I agree and have therefore add the suggested sentence. Effiom (ref 2) is a adequate reference as it explains the metastasizing amelobastoma, no need to add another ref.
- Line 145, Ideally serial follow up for the reported case should be at least 60 months and probably more up to 120. The authors should mention this in the article. Agree. I have added l. 148-149 a precision regarding this comment.
- 4. Line 317, replace the word indeed with "generally" the article reports cases of metastatic ameloblastoma which we could accept as not benign: OK done
- Comments on the Quality of English Language
- I have accepted and changed all your suggestions. Thank you for the comment
Reviewer 2 Report
Comments and Suggestions for Authors
The manuscript "ANTI-MAPK TARGETED THERAPY FOR AMELOBLASTOMA; CASE REPORT WITH A SYSTEMATIC REVIEW" describes the utility of Dabrafenib, a RAF inhibitor in the control of Ameloblastoma growth in a 15 year old female subject. This drug is utilized in the control of human cancers including melanoma carrying BRAF V600E mutation. Thus in this single subject trial, they could observe inhibition of cancer growth with the drug treatment for 2 months. To show that this drug might be useful for the treatment of Ameloblastoma, they have reviewed 13 studies of Ameloblastoma treated with dabrafenib in the presence and absence of other MAPK inhibitors.
The article is well written and the review of previous studies are well documented. The limitation is the single subject studied in this investigation and limited number of subjects from the literature. Authors have addressed these limitations indicating the possibility of selection bias and different treatment modalities for the limited number of cases from the literature. Overall, it is an interesting article for follow up studies on Ameloblastoma.
Author Response
Dear Reviewer,
Thank you very much for your positive comment on the article and your time taken. Indeed, there are few limitations (retrospective cases, number of cases) for this article, also because ameloblastoma remains a rare disease. Hopefully sharing this article might generate new research.
Thank you very much.
Reviewer 3 Report
Comments and Suggestions for Authors
1.In the case presentation part, OPG appeared without the full names(line 93).
Please specify the full names of OPG.
2.Line 102-103 said “at the time comprised an odontogenic keratocyst, keratocystic odontogenic tumour,”.
But odontogenic keratocyst and keratocystic odontogenic tumour is the same disease.
3.Line 105 it should have space between “(CBCT)were”.
4.Fig 1 b c appears after fig 2 in the manuscript.
Please present the figs in order.
5.Tables should have titles to summarize the main content.
Table 2 and 3 also need illustrations for the abbreviations.
6.Line 435-348 please add the corresponding reference. (Gryn-berg et al. ); Line 354-357 need the reference too.
7.In the table1, all the cases were checked for BRAF V600E mutation by biopsy? Please specify.
8.Institutional Review Board Statement.
No Institutional Review Board or number was seen.
9.There should be scale bars for the Fig 2.
10.Please specify there is no consensus for the following time of ameloblastoma after ANTI-MAPK TARGETED THERAPY.nTherefore, long time effects should be studied in the future.
Author Response
Dear Reviewer,
thank you very much for your relevant review and your time taken to review my article. Please find the detailed responses below.
1.In the case presentation part, OPG appeared without the full names(line 93). Please specify the full names of OPG. indeed, i added l. 94 "a dental panoramic radiograph". I have changed the OPG for panoramic radiograph (l. 99+102), thank you for pointing this out.
2.Line 102-103 said “at the time comprised an odontogenic keratocyst, keratocystic odontogenic tumour,”. But odontogenic keratocyst and keratocystic odontogenic tumour is the same disease. Indeed, it has been removed.
3.Line 105 it should have space between “(CBCT)were”. ok done
4.Fig 1 b c appears after fig 2 in the manuscript. Please present the figs in order. This is correct, as fig. 1 is radiographic evaluation thus fig. 1 b-c are post treatment and figure after fig 2 which is histological evaluation and must appear before "post treatment".
5.Tables should have titles to summarize the main content. Thank you for pointing this out. I have added a title.
6.Line 435-348 please add the corresponding reference. (Gryn-berg et al. ); Line 354-357 need the reference too. OK done
7.In the table1, all the cases were checked for BRAF V600E mutation by biopsy? Please specify. The mutational status has been specified either by PCR, IHC or NGS, which figure in the table.
8.Institutional Review Board Statement. No Institutional Review Board or number was seen. It has been added, thank you.
9.There should be scale bars for the Fig 2. Unfortunately we can not add a scale bar. The magnification is mentionned in the description below (either 20x, either 40x) for each slice and should be enough.
10.Please specify there is no consensus for the following time of ameloblastoma after ANTI-MAPK TARGETED THERAPY. Therefore, long time effects should be studied in the future. Agree. I have, accordingly, added a sentence to emphasize this interesting point (l. 394-395)